# Increased Apoptotic Activity in Low-Risk Myelodysplastic Syndrome

**DOI:** 10.3390/jcm11154604

**Published:** 2022-08-07

**Authors:** Songyi Park, Dong-Yeop Shin, Junseo Steve Park, Hee Sue Park, Soo Young Moon, Sung-Soo Yoon, Dong-Soon Lee

**Affiliations:** 1Division of Hematology/Oncology, Department of Internal Medicine, Kangbuk Samgung Hospital, Sungkyunkwan University School of Medicine, Seoul 03181, Korea; 2Division of Hematology and Medical Oncology, Department of Internal Medicine, Seoul National University Hospital, Seoul 03080, Korea; 3Biomedical Research Institute, Seoul National University Hospital, Seoul 03080, Korea; 4Cancer Research Institute, Seoul National University College of Medicine, Seoul 03080, Korea; 5Economics and Data Science Major, University of California, Berkeley, CA 94704, USA; 6Department of Laboratory Medicine, Chungbuk National University Hospital, Cheongju 28644, Korea; 7Department of Laboratory Medicine, Dongguk University Ilsan Hospital, Goyang-si 10326, Korea; 8Department of Laboratory Medicine, Seoul National University Hospital, Seoul 03080, Korea

**Keywords:** myelodysplastic syndrome, apoptosis, *TET2* mutation

## Abstract

Myelodysplastic syndrome (MDS) is a heterogeneous hematopoietic disorder associated with cellular proliferative and apoptotic activity. We retrospectively investigated these activities in bone marrow samples from 76 MDS patients using immunohistochemical staining for Ki-67 and cleaved caspase-3. We divided cleaved caspase-3 into two groups based on median value and compared the differences according to MDS risk scoring systems. We compared MDS patient indices with idiopathic cytopenia of undetermined significance (ICUS) and healthy control (HC) indices using our previously published data. Cleaved caspase-3 immunohistochemistry was highest in MDS patients, followed by ICUS patients and HCs. Similarly, the mean Ki-67 grade was also highest in MDS patients, followed by ICUS patients and HCs. Higher cleaved caspase-3 grade was significantly associated with lower IPSS-R score (*p* = 0.020), whereas Ki-67 was not associated with MDS. Interestingly, *TET2* mutation was associated with decreased cleaved caspase-3 levels (*p* = 0.03). However, there was no significant association between proliferative/apoptotic activity and survival. Our results suggest that apoptotic activity gradually increases from healthy controls and ICUS patients to MDS patients. Furthermore, higher apoptotic activity was associated with better MDS patient prognostic scores. Further studies are needed to reveal the differences in apoptotic activity between lower- and higher-risk MDS.

## 1. Introduction

Myelodysplastic syndrome (MDS) is a clonal hematopoietic stem cell disorder characterized by altered proliferation and differentiation. Patients with MDS usually show hypercellular marrow with cellular dysplasia, resulting in anemia and other cytopenias [1]. The major clinical problems in MDS patients include morbidities caused by cytopenia and the potential evolution to acute myeloid leukemia. Characteristic chromosomal abnormalities, such as del (20q), del (5q), and −7, and recurrent genetic mutations in epigenetic modifiers and splicing genes have been identified as MDS-causing mutations [2,3]. Although the cause and pathological process of MDS development is unclear, with the exception of known genetics, increasing evidence of disease mechanisms has accumulated [1,4].

Short telomere length (TL) could be an independent prognostic marker in MDS patients, but not in acute myeloid leukemia patients, which suggests a unique pathophysiology of MDS [3]. MDS might progress through idiopathic cytopenia of undetermined significance (ICUS), considering previous studies showing gradual shortening of TL in healthy controls (HCs) and ICUS patient [5,6]. These findings indicate that MDS is associated with increased proliferative activity resulting in telomere erosion and senescence of marrow cells [7].

Along with the progressive telomere shortening, cellular proliferative capacity decreases, and apoptosis rate increases. There is morphological evidence of increased apoptosis of erythroid and immature myeloid precursors in bone marrow (BM) biopsies from MDS patients [8]. Apoptosis in the BM microenvironment is regulated by numerous factors, including oncogenes, growth factors, and immunological factors. Some studies have provided evidence that increased erythroid apoptosis is relevant, especially in MDS, and is associated with mitochondrial dysfunction, resulting in the activation of effector caspases and increased sensitivity to death ligands [9].

Caspase-3 is a pro-apoptotic protease involved in mediating cell death signaling transduction [10]. Caspase-3 is an effector caspase that is activated and induces proteolytic cleavage leading to cell death. In contrast, the Ki-67 index of BM biopsy is known to be a proliferation marker. Previous studies have proposed that Ki-67 expression could explain the difference in biological behavior between high- and low-grade MDS [11].

Therefore, we investigated whether there was an association between caspase-3/Ki-67 expression and prognostic scores in patients with MDS. Specifically, proliferative and apoptotic activity in MDS patients was compared to that in HCs and ICUS patients, and the role of these activities was explored according to prognostic risk scores.

## 2. Materials and Methods

### 2.1. Study Design and Population

This was a retrospective study that used primary BM-derived blood samples. Eligible patients were identified from the electronic database, and their medical charts were reviewed from the electronic database medical record system of Seoul National University Hospital (SNUH, Seoul, Korea). Patients who were initially diagnosed with MDS between 2004 and 2014 at SNUH were included. Patients were excluded if primary BM samples for immunohistochemical (IHC) staining at the time of initial diagnosis were not available.

The study protocol was reviewed and approved by the institutional review board of our institution. All procedures, including the study treatment, follow-up, and data collection, were conducted in accordance with the Declaration of Helsinki.

### 2.2. IHC

IHC staining for cleaved caspase-3 and Ki-67 was performed on the BM biopsy sections. Paraffin-embedded tissue blocks were divided into 2 µm sections, and each slide was stained using the Ventana BenchMark ULTRA automated staining platform (Ventana Medical Systems Inc., Tucson, AZ, USA). Cleaved caspase-3 (1:100; Cell Signaling Technology, Danvers, MA, USA) and Ki-67 (1:100; DAKO, Glostrup, Denmark) monoclonal antibodies were applied for 15 min at room temperature. Two expert hematopathologists independently reviewed the stained slides and measured the percentage of cells positive for cleaved caspase-3 and Ki-67. We converted the percentage into grades on a 4-grade scale [5] and analyzed the results using the average of the two expert hematopathologist values. Clinical information of ICUS and HCs, which had been published elsewhere [5], was used to compare the degrees of Ki-67 and caspase-3 expression with MDS in this study.

### 2.3. Telomere Q-FISH

Cryopreserved samples were used for telomere analysis by quantitative fluorescence in situ hybridization (Q-FISH). Q-FISH was performed using a Cy3-labeled telomere peptide nucleic acid (PNA) FISH kit (DakoCytomation Denmark A/S, Glostrup, Denmark) and a fluorescein isothiocyanate (FITC)-labeled PNA probe for the centromere of chromosome 2 (kindly provided by DakoCytomation).

### 2.4. Prognostic Factors

We divided MDS patients into two groups based on the median value of cleaved caspase-3. We also compared the basic patient characteristics between the groups. The international prognostic scoring system (IPSS) and revised IPSS (IPSS-R) for MDS patients were used in order to investigate whether Ki-67 and cleaved caspase-3 can be used as prognostic factors [12]. We analyzed the association between the prognostic scores and Ki-67 and cleaved caspase-3 levels.

### 2.5. Mutational Analysis

Targeted sequencing of 87 hematologic malignancy-associated genes was performed using next-generation sequencing on an Illumina HiSeq 2500 platform (Illumina, San Diego, CA, USA).

### 2.6. Statistics

Statistical analyses of clinical parameters were performed using IBM SPSS Statistics version 25 (SPSS, Chicago, IL, USA). The chi-square test and logistic regression analysis were used to investigate the association between Ki-67, cleaved caspase-3, and clinical variables. We divided groups according to Ki-67, cleaved caspase-3, IPSS and IPSS-R, and we used *t*-test and one-way ANOVA to compare the differences between groups. The Kaplan–Meier method was used to obtain estimates of median overall survival (OS) and progression-free survival (PFS), and comparisons were made using log-rank tests. Two-sided *p* values < 0.05 were considered statistically significant.

## 3. Results

### 3.1. Patient Characteristics

A total of 76 MDS patients were enrolled in this study. The median age of the patients was 70 years (range, 22–86). Fifty-four patients (71.1%) were male. The baseline characteristics, including clinical information, are described in Table 1. We divided the patients into two groups based on the median cleaved caspase-3 value. The high cleaved caspase-3 level group had more female patients and showed less thrombocytopenia than the low cleaved caspase-3 level group (Table 1).

### 3.2. Comparison of Cleaved Caspase-3, Ki-67 Grade, and TL among MDS Patients, ICUS, and HCs

Data on cleaved caspase-3, Ki-67, and TL in patients with ICUS (*n* = 37) and HCs (*n* = 49), which had been published elsewhere [5] were used for comparison with MDS patients. The mean cleaved caspase-3 grade was highest in MDS patients, followed by ICUS patients and HCs (2.70, 95% CI: 2.56–2.85 vs. 2.19, 95% CI: 1.85–2.54 vs. 1.45, 95% CI: 1.17–1.73, *p* < 0.001, respectively) (Figure 1A). Similarly, the mean Ki-67 grade was also the highest in MDS patients, followed by ICUS patients and HCs (1.72, 95% CI: 1.55–1.90 vs. 1.58, 95% CI: 1.30–1.86 vs. 1.72, 95% CI: 0.95–1.16, *p* = 0.001, respectively) (Figure 1B). TL was significantly shorter in patients with MDS than in HCs, but similar to that in patients with ICUS (Figure 1C). Interestingly, cleaved caspase-3 grade gradually increased in HCs, ICUS, and MDS.

### 3.3. Higher Cleaved Caspase-3 Grade Was Enriched in Lower-Risk MDS

We divided cleaved caspase-3 and Ki-67 values into two groups based on their median values and analyzed associations with the IPSS-R score in MDS patients. A higher cleaved caspase-3 grade was significantly associated with a lower IPSS-R score (*p* = 0.020) (Figure 2A). However, there was no correlation between the Ki-67 grade and IPSS-R score (*p* = 0.976) (Figure 2B). We then divided MDS patients into two groups according to the IPSS-R risk stratification: very low to intermediate risk in low-risk MDS and high to very high in high-risk MDS. We also compared cleaved caspase-3 and Ki-67 grades among patients with ICUS, low-risk MDS, and high-risk MDS. Cleaved caspase-3 levels in patients with low-risk MDS were significantly higher, followed by those in patients with high-risk MDS and ICUS (*p* = 0.004, ICUS vs. low-risk MDS, *p* = 0.009; low-risk MDS vs. high-risk MDS, *p* = 0.779; ICUS vs. high-risk MDS, *p* = 0.044) (Figure 3A). Ki-67 values tended to increase in the order of patients with ICUS, low-risk MDS, and high-risk MDS, but the difference was not statistically significant (Figure 3B). We also analyzed cleaved caspase-3 and Ki-67 levels over stratified IPSS-R scores in patients with MDS, but the results were not statistically significant (Figure 4A,B).

### 3.4. Ki-67, Caspase-3, and Genetics in MDS

The most common genetic abnormality in patients with MDS was *ASXL1* (*n* = 17), followed by *U2AF1* (*n* = 13), and *TP53* (*n* = 11) (Table 2). In general, there were no significant differences in age, Ki-67 index, or TL. No significant gene mutations were associated with TL or Ki-67 grade. However, patients with *TET2* mutations showed significantly lower levels of cleaved caspase-3 than those without *TET2* mutations (Table 2).

### 3.5. Outcomes

We analyzed whether TL, cleaved caspase-3, and Ki-67 were associated with treatment response. TL in MDS patients who responded to treatment was shorter than that in MDS patients who did not respond to treatment (*p* = 0.015) (Figure 5A). However, there were no significant differences in cleaved caspase-3 and Ki-67 values according to treatment response (Figure 5B,C). PFS and OS were analyzed according to cleaved caspase-3 and Ki-67 grades, but there were no significant findings (Figure 6A,B and Figure 7A,B).

In addition, we divided patients with MDS into four groups: lower-risk MDS patients with lower caspase-3 levels (*n* = 32), lower-risk MDS patients with higher caspase-3 levels (*n* = 13), higher-risk MDS patients with lower caspase-3 levels (*n* = 28), and higher-risk MDS patients with higher caspase-3 levels (*n* = 2). We compared PFS and OS between the four groups and found that high-risk MDS patients with lower caspase-3 showed significantly worse outcomes than the other groups, followed by low-risk MDS patients with higher caspase-3 and low-risk MDS patients with lower caspase-3 levels (Figure 8A,B).

## 4. Discussion

We assessed TL, cleaved caspase-3, and Ki-67 in patients with MDS and compared them with ICUS patients and HCs. The TL in MDS was the shortest and became longer in the order of ICUS patients and HCs. Both cleaved caspase-3 and Ki-67 showed high values in MDS and decreased values in ICUS patients and HCs. These findings suggest that MDS patients show higher apoptotic activity and higher proliferative activity than HCs or ICUS patients.

Although the evolution of cancer is complex and still unknown, previous studies have suggested that deregulated cell proliferation and apoptosis could be the cause of neoplastic progression. Previous studies have suggested that clonal expansion of leukemic cells is due to proliferation in excessive apoptosis, or that oncogenic proliferation can result from a variety of growth-related processes, such as apoptosis, differentiation, or senescence [13,14]. Increased activities of pro-apoptotic proteins, such as caspases, can induce proliferation of neighboring surviving cells and are related to a higher grade of tumor malignancy and poorer clinical outcomes [13,15]. Previous suggestions that proliferation and apoptosis increase together with cancer progression are consistent with our findings showing their increase along with MDS progression.

An increased apoptotic signal has been associated with poor prognosis in patients with various solid tumors [14,16]. Apoptotic cancer cells with elevated cleaved caspase-3 levels use their apoptotic signals to generate potent growth-stimulating fuels to stimulate tumor repopulation [17,18]. Apoptotic signals, even in normal cells residing in the tumor microenvironment, can stimulate malignant cells via a caspase 3-mediated pathway [19].

Higher levels of cleaved caspase-3 were more frequently observed in lower IPSS-R scores (Figure 2A). As a result, cleaved caspase-3 gradually increased from HCs and ICUS to lower-risk MDS patients, and then showed a tendency to decrease in higher-risk MDS patients. Based on previous and current reports, our results suggest that apoptotic stimuli increase from the ICUS stage, and then trigger malignant transformation via a caspase-3-related pathway at the lower-risk MDS stage. However, progression from lower-risk MDS to higher-risk MDS might not be apoptosis-driven leukemogenesis, considering our observations showing a lack of increase in cleaved caspase-3 level. Our results suggest that lower-risk and higher-risk MDS might be discriminated into two disease subgroups with different pathophysiologies. Thus, we suggest that cleaved caspase-3 may be a prognostic risk factor in patients with MDS.

Furthermore, our results showed significantly poor outcomes in patients with higher risk of IPSS-R and lower cleaved caspase-3 levels (Figure 8A,B). With further study, the potential prognostic role of cleaved caspase-3 could be revealed.

Although there was a significant association between cleaved caspase-3 grade and IPSS-R score, we could not find a statistically significant association between apoptosis markers and IPSS risk score (Appendix A). IPSS-R was developed to evaluate cytogenetic characteristics and cytopenia in more details, compared with IPSS, and previous studies have shown that it is more accurate for clinical prognosis than IPSS [20,21]. IPSS-R is a more appropriate prognostic scoring system to predict clinical outcomes in the real world, and, consistently, meaningful results were derived from our study as well.

There is little evidence of an association between genetic abnormalities and apoptotic activity in patients with MDS. In this study, we performed genetic analysis and compared apoptotic and proliferative activities according to genetic mutations. We found no significant association between gene mutations and Ki-67 or TL. However, patients with *TET2* mutations showed lower levels of cleaved caspase-3 than those without mutations. *TET2* is one of the most frequently mutated genes in myeloid neoplasms, including MDS and chronic myelomonocytic leukemia, and is a negative prognostic factor [22]. Our finding that decreased cleaved caspase-3 is related to a higher risk of MDS is consistent with the negative prognostic role of *TET2* mutation. Further studies on the prognosis of patients with MDS and their genetic mutations are needed.

Our study is limited by the small number of enrolled patients and the retrospective nature of the study. However, our study still has value, because our findings could provide an insight into disease progression from ICUS to MDS and the difference between lower- and higher-risk MDS. Further studies are warranted to investigate the different pathophysiologies of lower-risk and higher-risk MDS.

In summary, apoptotic activity measured through cleaved caspase-3 level increases in the order of HCs, ICUS patients, and MDS patients. In patients with MDS, increased levels of cleaved caspase-3 were consistently observed in patients with lower-risk MDS determined using IPSS-R, which stopped increasing and rather slightly decreased in patients with higher-risk MDS. In addition, *TET2* mutations were associated with decreased cleaved caspase-3 activity.

## Figures and Tables

**Figure 1 jcm-11-04604-f001:**
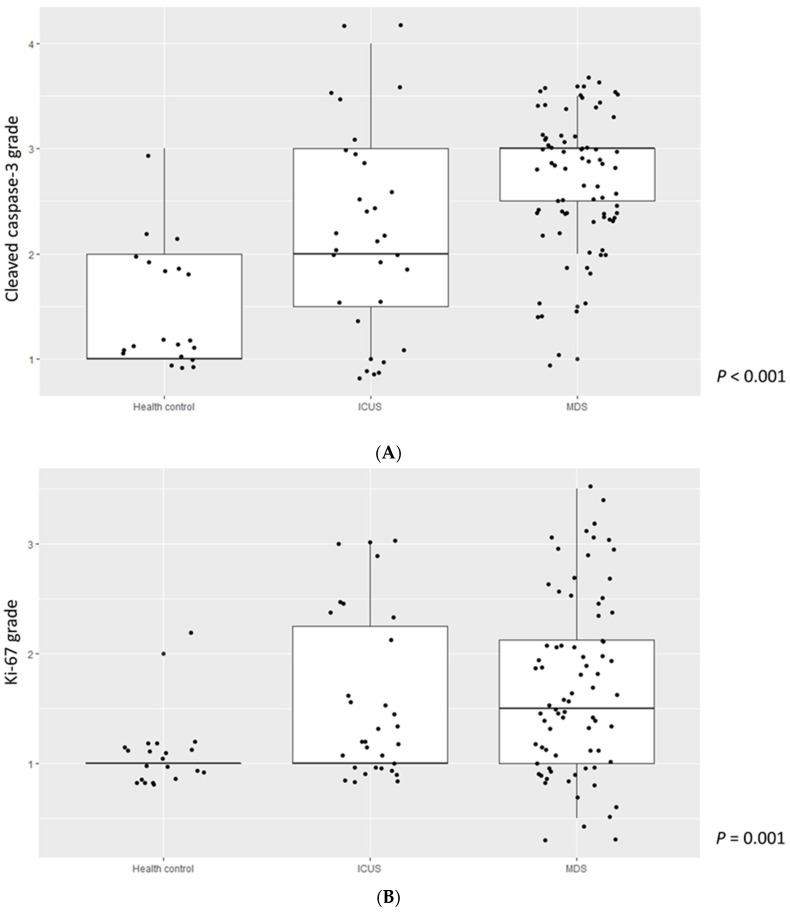
(**A**) Cleave caspase-3 grade in patients with MDS, ICUS, HCs. Abbreviation: MDS, myelodysplastic syndrome; ICUS, Idiopathic cytopenia of undetermined significance; HCs, Health controls. (**B**) Ki-67 grade in patients with MDS, ICUS, HCs. Abbreviation: MDS, myelodysplastic syndrome; ICUS, Idiopathic cytopenia of undetermined significance; HCs, Health controls. (**C**) Telomere length in patients with MDS, ICUS, HCs. Abbreviation: MDS, myelodysplastic syndrome; ICUS, Idiopathic cytopenia of undetermined significance; HCs, Health controls.

**Figure 2 jcm-11-04604-f002:**
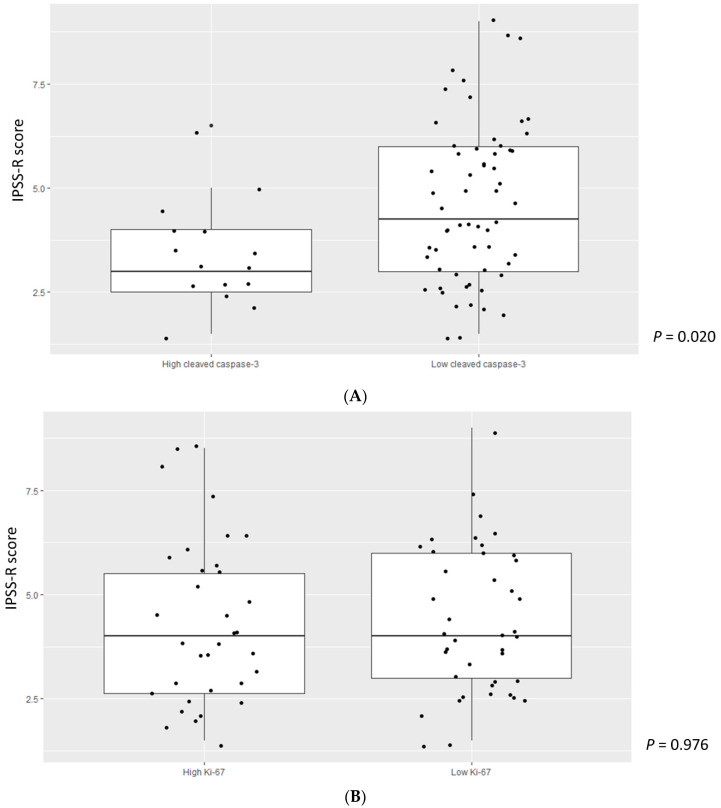
(**A**) IPSS-R score according to cleaved caspase-3. Abbreviation: IPSS-R, Revised international prognostic scoring system. (**B**) IPSS-R score according to Ki-67. Abbreviation: IPSS-R, Revised international prognostic scoring system1.

**Figure 3 jcm-11-04604-f003:**
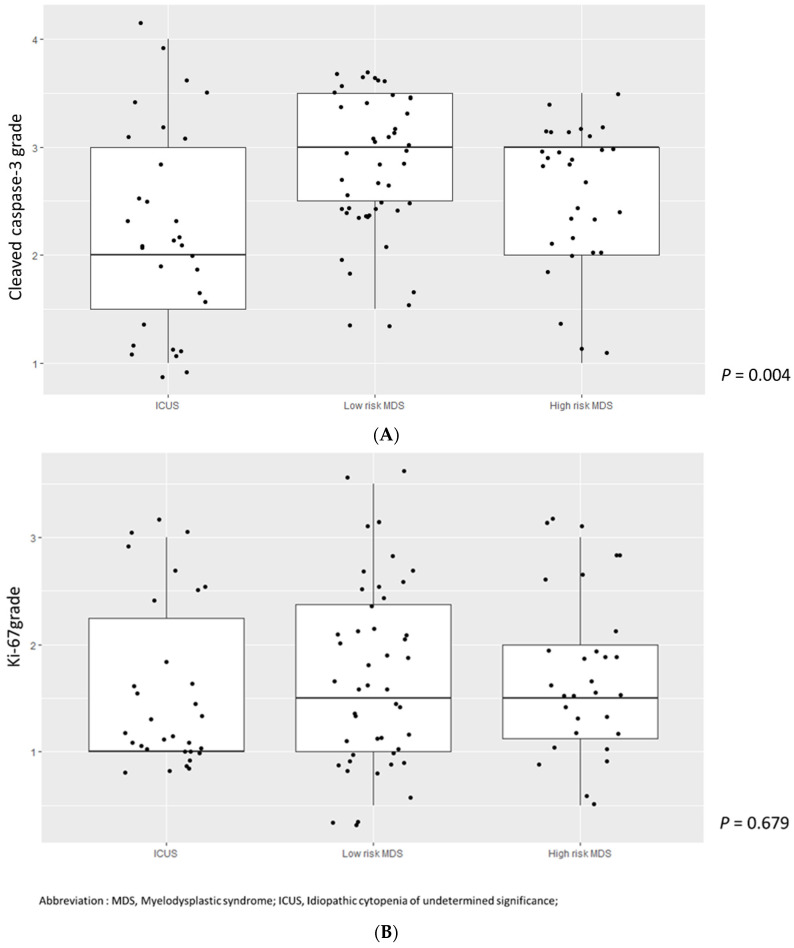
(**A**) Cleaved caspase-3 grade in patients with ICUS, low risk MDS and high risk MDS. Abbreviation: MDS, Myelodysplastic syndrome; ICUS, Idiopathic cytopenia of undetermined significance. (**B**). Ki-67 grade in patients with ICUS, low risk MDS and high risk MDS. Abbreviation: MDS, Myelodysplastic syndrome; ICUS, Idiopathic cytopenia of undetermined significance.

**Figure 4 jcm-11-04604-f004:**
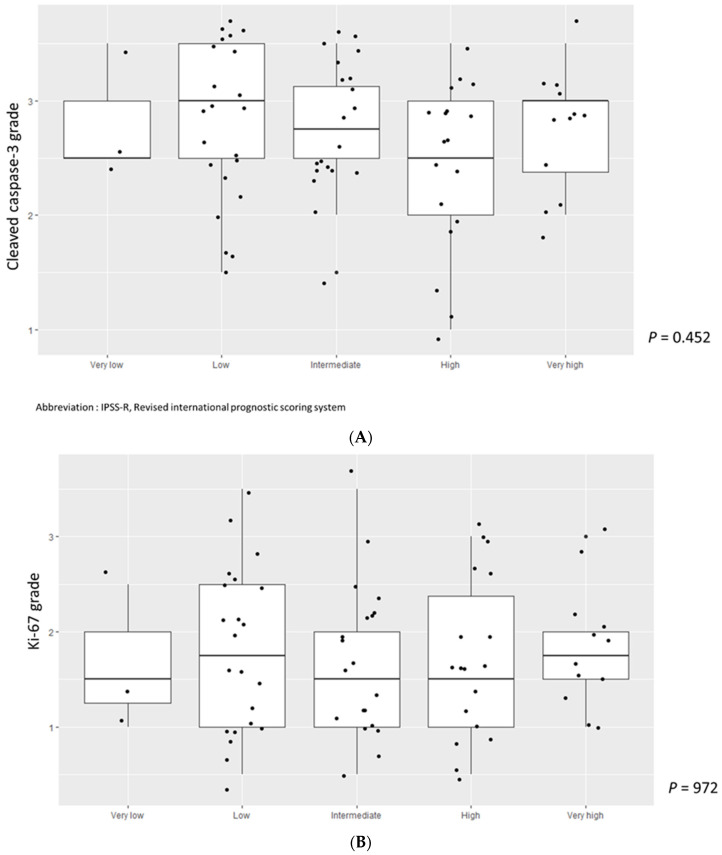
(**A**) Cleaved caspase-3 grade according to IPSS-R risk stratification. Abbreviation: IPSS-R, Revised international prognostic scoring system. (**B**) Ki-67 grade according to IPSS risk stratification. Abbreviation: IPSS-R, Revised international prognostic scoring system.

**Figure 5 jcm-11-04604-f005:**
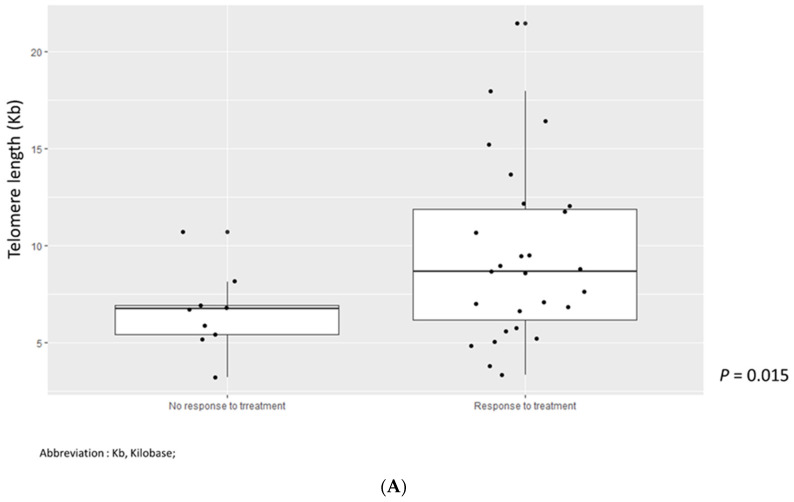
(**A**) Telomere length according to response to treatment. Abbreviation: Kb, kilobase. (**B**) Cleaved caspase-3 grade according to response to treatment. (**C**) Ki-67 grade according to response to treatment.

**Figure 6 jcm-11-04604-f006:**
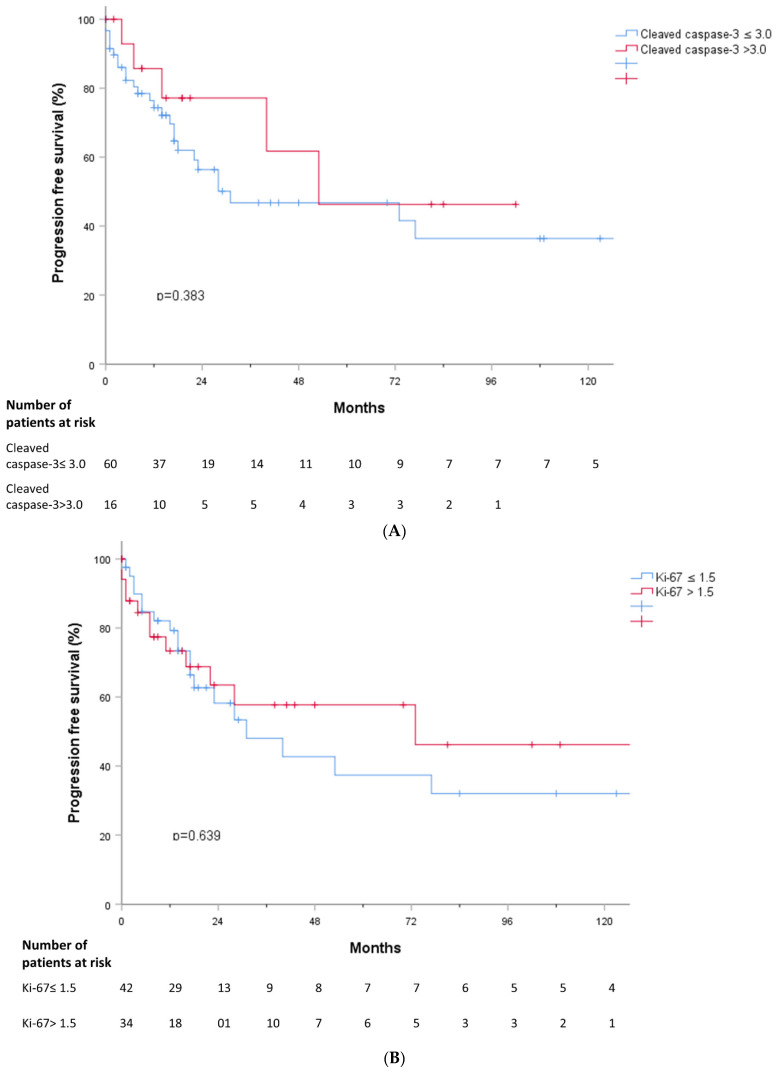
(**A**) Progression free survival according to cleaved caspase-3. (**B**) Progression free survival according to Ki-67.

**Figure 7 jcm-11-04604-f007:**
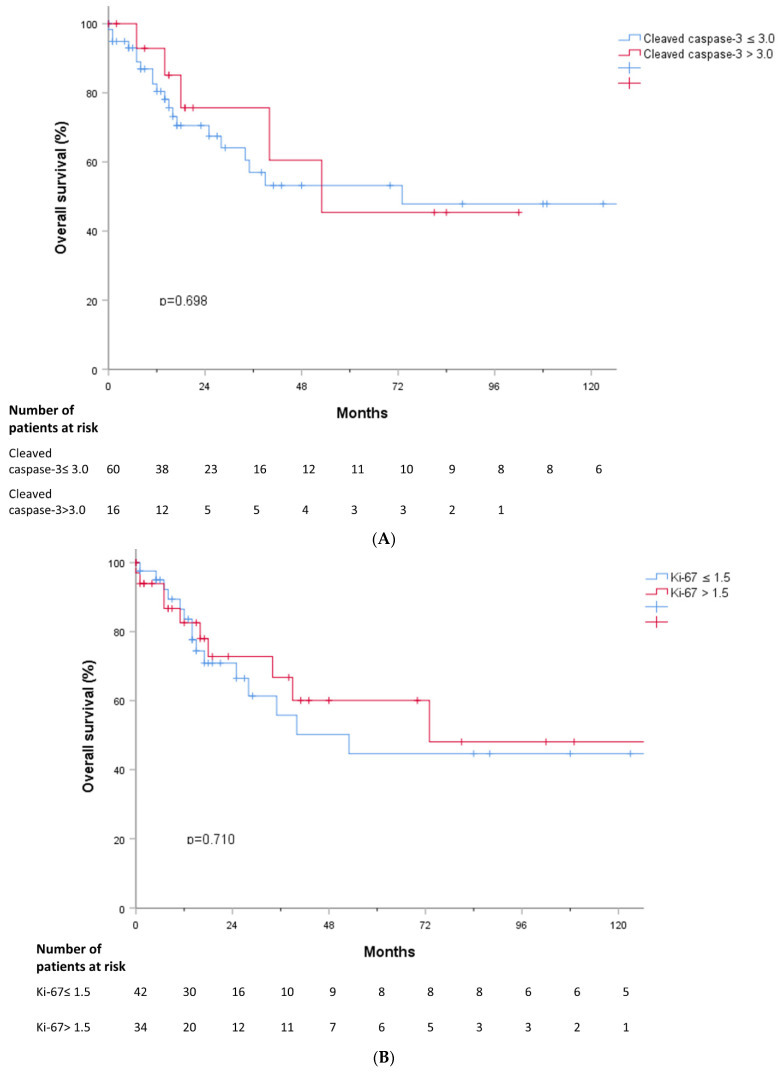
(**A**) Overall survival according to cleaved caspase-3. (**B**) Overall survival according to Ki-67.

**Figure 8 jcm-11-04604-f008:**
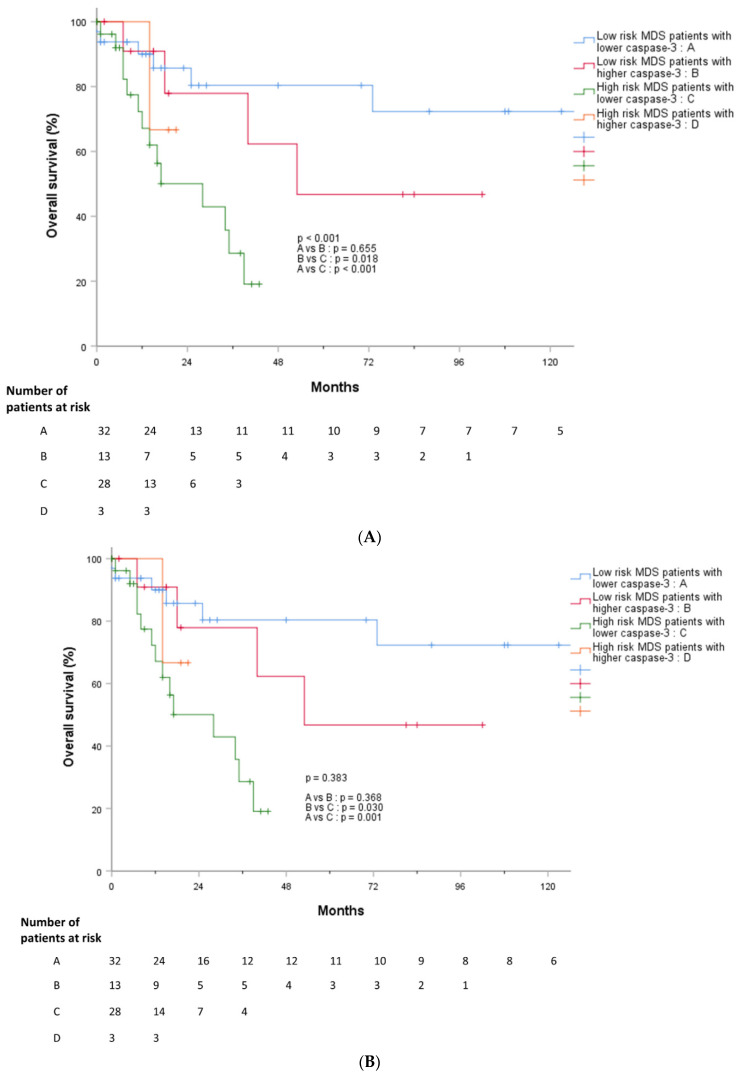
(**A**) Overall survival according to cleaved caspase-3 and IPSS-R risk stratification. Abbreviation: IPSS-R, Revised international prognostic scoring system; MDS, Myelodysplastic syndrome; (**B**) Overall survival according to Ki-67 and IPSS-R risk stratification. Abbreviation: IPSS-R, Revised international prognostic scoring system; MDS, Myelodysplastic syndrome.

**Table 1 jcm-11-04604-t001:** Patient characteristics.

Variables	Total (*n* = 76)	Low Cleaved Caspase-3 (*n* = 60, 78.9%)	High Cleaved Caspase-3 (*n* = 16, 21.1%)	*p*-Value
Age at diagnosis, median (range)	70 (22–86)	70 (25–86)	72 (22–86)	0.656
Sex, *n* (%)				0.037
Male Female	54 (71.1) 22 (28.9)	46 (76.7) 14 (23.3)	8 (50.0) 8 (50.0)	
BM, median (range)				
Cellularity, % Blast at diagnosis, %	55 (5–95) 2.40 (0.0–19.1)	55 (5–95) 2.40 (0.0–19.1)	45 (15–95) 2.75 (0.0–19.0)	0.223 0.463
Cytopenia, *n* (%)				
Neutropenia (ANC < 1800/mm^3^)	52 (68.4)	40 (66.7)	12 (75.0)	0.524
Anemia (Hb < 10 g/dL)	59 (77.6)	47 (78.3)	12 (75.0)	0.776
Thrombocytopenia (Platelet <100,000/mm^3^)	46 (60.5)	40 (66.7)	6 (37.5)	0.034
Lineage involvement of cytopenia, *n* (%)				0.816
Uni-lineage	26 (34.2)	21 (35.0)	5 (31.3)	
Bi-lineage	27 (35.5)	22 (36.7)	5 (31.3)	
Tri-lineage	17 (22.4)	12 (20.0)	5 (31.3)	
Cytopenia grade, *n* (%)				0.513
1	19 (25.0)	15 (25.0)	4 (25.0)	
2	24 (31.6)	17 (28.3)	7 (43.8)	
3	32 (39.5)	26 (86.7)	4 (25.0)	
IPSS risk, *n* (%)				0.082
Low	10 (13.2)	6 (10.0)	4 (25.0)	
Intermediate-1	43 (56.6)	34 (56.7)	9 (56.3)	
Intermediate-2	17 (22.4)	16 (26.7)	1 (6.3)	
High	5 (6.6)	4 (6.7)	1 (6.3)	
Revised IPSS risk, *n* (%)				0.090
Very low	3 (3.9)	2 (3.3)	1 (6.3)	
Low	22 (28.9)	15 (25.0)	7 (43.8)	
Intermediate	20 (26.3)	15 (25.0)	5 (31.3)	
High	18 (23.7)	17 (28.3)	1 (6.3)	
Very high	12 (15.8)	11 (18.3)	1 (6.3)	
Karyotype, *n* (%)				0.086
Abnormal karyotype	38 (50.0)	18 (30.0)	7 (43.8)	
Complex karyotype (>3 abnormalities)	13 (17.1)	13 (21.7)	0 (0.0)	
Del(5q)/−5	7 (9.2)	6 (10.0)	1 (6.25)	0.559
Del(7q)/−7	12 (15.8)	11 (18.3)	1 (6.25)	0.179
Trisomy 8	12 (15.8)	10 (16.7)	2 (12.5)	0.539
Del(20q)	6 (7.9)	5 (8.3)	1 (6.25)	0.770
Telomere length, mean kb (range)	8.65 (3.2–21.47)	8.35 (3.2–19.04)	10.77 (5.74–21.47)	0.169
Ki-67, mean grade (range)	1.5 (0.5–3.5)	1.5 (0.5–3.5)	1.75 (0.5–2.5)	0.700

Abbreviation: BM, bone marrow; IPSS, International prognostic scoring system; ANC, Absolute neutrophil count.

**Table 2 jcm-11-04604-t002:** Mutational profiles.

Gene Mutation		Mean Age (Years)	*p*-Value	Mean Cleaved Caspase-3 (Grade)	*p*-Value	Mean Ki-67 (Grade)	*p*-Value	Mean Telomere Length (kb)	*p*-Value
Any mutation	+ (*n* = 60)	71 (27–86)	0.165	3.0 (1.0–3.5)	0.294	1.5 (0.5–3.5)	0.109	8.73 (3.33–21.47)	0.914
	− (*n* = 16)	68.5 (22–86)		2.5 (1.5–3.5)		1.75 (1.0–3.0)		6.99 (3.20–17.79)	
*ASXL1*	+ (*n* = 17)	72 (46–86)	0.065	3.0 (1.0–3.5)	0.680	2.0 (1.0–3.5)	0.194	8.65 (3.33–2147)	0.963
	− (*n* = 59)	70 (22–86)		3.0 (1.5–3.5)		1.5 (0.5–3.0)		8.68 (3.20–19.04)	
*TP53*	+ (*n* = 11)	71 (61–84)	0.101	3.0 (1.0–3.5)	0.897	2.0 (1.0–3.0)	0.398	10.65 (3.33–16.0)	0.914
	− (*n* = 65)	70 (22–86)		3.0 (1.0–3.5)		1.5 (0.5–3.5)		8.61 (3.20–21.47)	
*EZH2*	+ (*n* = 5)	72 (64–86)	0.274	3.0 (1.0–3.5)	0.730	1.0 (1.0–2.5)	0.511	11.75 (8.35–21.47)	0.081
	− (*n* = 71)	70 (22–86)		3.0(1.0–3.5)		1.5 (0.5–3.5)		8.62 (3.20–19.04)	
*U2AF1*	+ (*n* = 13)	60 (27–84)	0.057	3.0 (1.5–3.5)	0.688	1.0 (0.5–3.0)	0.259	8.80 (3.79–19.04)	0.989
	− (*n* = 63)	71 (22–86)		3.0 (1.0–3.5)		1.5 (0.5–3.5)		8.35 (3.20–21.47)	
*RUNX1*	+ (*n* = 5)	64 (33–78)	0.223	2.5 (1.5–3.0)	0.145	1.5 (0.5–3.0)	0.717	8.91 (5.03–17.96)	0.794
	− (*n* = 71)	71 (22–86)		3.0 (1.0–3.5)		1.5 (0.5–3.5)		8.65 (3.20–21.47)	
*TET2*	+ (*n* = 10)	74 (67–86)	0.030	2.25 (1.0–3.5)	0.031	1.75 (0.5–3.0)	0.751	11.0 (8.14–16.22)	0.228
	− (*n* = 66)	70 (22–86)		3.0 (1.0–3.5)		1.5 (0.5–3.5)		8.35 (3.20–21.47)	
*DNMT3A*	+ (*n* = 6)	76 (56–86)	0.186	3.0 (2.5–3.0)	0.399	1.25 (0.5–2.0)	0.122	8.14 (4.83–9.97)	0.343
	− (*n* = 70)	70 (22–86)		2.75 (1.0–3.5)		1.5 (0.5–3.5)		8.67 (3.20–21.47)	
*SRSF2*	+ (*n* = 6)	72 (65–86)	0.186	3.0 (1.0–3.5)	0.140	1.5 (0.5–2.0)	0.204	9.08 (5.33–19.04)	0.284
	− (*n* = 70)	70 (22–86)		2.5 (1.5–3.0)		1.5 (0.5–3.5)		8.58 (3.20–21.47)	
*BCOR*	+ (*n* = 3)	68 (47–83)	0.982	3.0 (1.0–3.5)	0.576	2.0 (0.5–3.0)	0.806	8.78 (2.19–17.96)	0.685
	− (*n* = 73)	70 (22–86)		2.5 (2.0–3.0)		1.5 (0.5–3.5)		8.62 (3.20–21.47)	
*SF3B1*	+ (*n* = 5)	70 (46–73)	0.739	2.5 (1.5–3.5)	0.464	2.5 (1.0–3.5)	0.159	9.97 (5.84–14.16)	0.909
	− (*n* = 71)	71 (22–86)		3.0 (1.0–3.5)		1.25 (0.5–3.5)		8.62 (3.20–21.47)	
*STAG2*	+ (*n* = 4)	68 (27–75)	0.395	2.75 (2.0–3.5)	0.883	1.75 (1.5–2.0)	0.945	8.69 (5.59–18.00)	0.651
	− (*n* = 72)	70 (22–86)		3.0 (1.0–3.5)		1.5 (0.5–3.5)		8.62 (3.20–21.47)	
*WT1*	+ (*n* = 4)	60 (43–67)	0.262	2.5 (1.5–3.5)	0.515	1.5 (1.0–2.5)	0.797	7.81 (3.79–8.69)	0.206
	− (*n* = 72)	71 (22–86)		3.0 (1.0–3.5)		1.5 (0.5–3.5)		8.77 (3.20–21.47)	
Chromosomal abnormality		Mean age (years)	*p*-value	Mean cleaved caspase-3 (grade)	*p*-value	Mean Ki-67 (grade)	*p*-value	Mean telomere length (kb)	*p*-value
IPSS									
Good	*n* = 44 (57.9%)	70.5 (22–86)	0.240	2.5 (1.0–3.5)	0.919	1.5 (0.5–3.5)	0.541	8.67 (3.20–21.47)	0.960
Intermediate	*n* = 16 (21.1%)	70 (27–84)		3.0 (1.5–3.5)		1.75 (0.5–3.0)		7.64 (3.79–17.96)	
Poor	*n* = 15 (19.7%)	71 (55–83)		3.0 (1.0–3.5)		2.0 (0.5–3.0)		9.03 (3.33–16.0)	
IPSS-R									
Very good	*n* = 3 (3.9%)	61 (56–75)	0.564	3.5 (3.0–3.5)	0.404	2.0 (1.5–3.5)	0.508	9.41 (6.78–12.05)	0.680
Good	*n* = 43 (56.6%)	71 (22–86)		2.5 (1.0–3.5)		1.5 (0.5–3.5)		8.69 (3.20–21.47)	
Intermediate	*n* = 15 (19.7%)	70 (27–84)		3.0 (1.5–3.5)		1.5 (0.5–3.0)		7.64 (3.79–17.96)	
Poor	*n* = 13 (17.1%)	71 (55–83)		3.0 (1.0–3.5)		2.0 (0.5–3.0)		8.47 (3.33–16.00)	
Very poor	*n* = 1 (1.3%)	63		3.0		1.5		15.18	

Abbreviations: kb, kilobase; *AXL1*, Additional sex combs-like 1; *TP53*, Tumor protein p53; *EZH2*, Enhancer of zeste homolog 2; *U2AF1I,* U2 small nuclear RNA auxiliary factor 1; *RUNX1,* Runt-related transcription factor 1; *TET2,* Tet methylcytosine dioxygenase 2; DNMT3A, DNA methyltransferase 3A; *SRSF2,* Serine and arginine rich splicing factor 2; *BCOR,* BCL6 corepressor; *SF3B1,* Splicing factor 3b subunit 1; *STAG2*, Stromal antigen 2; *WT1*, wilms tumor-1; IPSS, International prognostic scoring system; IPSS-R, Revised international prognostic scoring system.

## Data Availability

Not applicable.

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
