# Peer review of "Increased Apoptotic Activity in Low-Risk Myelodysplastic Syndrome"

_jcm, 2022, doi:10.3390/jcm11154604_

Round 1

Reviewer 1 Report

The study by Park et al. describes the expression and clinical-laboratory correlation of cleaved caspase 3 expression in bone marrow samples from patients with MDS. The role of apoptosis in MDS has been extensively debated in recent decades, but there is still no clear conclusion on the topic. Thus, the study by Park and colleagues elegantly contributes to this body of evidence on the relationship between apoptosis and MDS. The study is well written and some points could be improved for publication:

1- Although authors report that data on ICUS and HC individuals are published in reference 5. Basic information such as inclusion criteria for ICUS, age (median, min-max), and gender need to be described in the current work, since this information impacts readers' interpretation.

2- Which statistical method was used to compare groups? Include material and methods and figure legends.

3- In graphs with more than two groups, which comparison does the P value refer to? Was a post-test between the groups carried out? This reviewer strongly suggests the application of the Kruskal-Walis and post-test Dunn's test.

4- Boxplot plots would be better presented together with the individual points included in the figure. This way the reader will have a clearer interpretation of the data distribution.

5 - In the survival curves, authors must include the number of "patients at risk" for each time interval. This improves readers' interpretation.

Author Response

1) Although authors report that data on ICUS and HC individuals are published in reference 5. Basic information such as inclusion criteria for ICUS, age (median, min-max), and gender need to be described in the current work, since this information impacts readers' interpretation.

Answers to 1) : Thank you for your delicate instruction. We reorganized the table from the previous published paper and attached as supplement table.

2) Which statistical method was used to compare groups? Include material and methods and figure legends.

Answers to 2) : Thanks for pointing out what we missed. We used t-test to compare two groups and we also applied one-way ANOVA to compare more than two groups. We additionally described about statistical methods in the text.

Page 3. Line 116-117 : And we divided groups according to Ki-67, cleaved caspase-3, IPSS and IPSS-R. And we used t-test and one-way ANOVA to compare the differences between groups.

3) In graphs with more than two groups, which comparison does the P value refer to? Was a post-test between the groups carried out? This reviewer strongly suggests the application of the Kruskal-Walis and post-test Dunn's test.

Answer to 3) : Thank you for your insightful comment. We absolutely agree with reviewer’s comments about further analysis. We additionally performed statistical analysis as you recommended. And it showed the similar results with the previous analysis. We described analysis results in the graph and tables below.

4) Boxplot plots would be better presented together with the individual points included in the figure. This way the reader will have a clearer interpretation of the data distribution.

Answers to 4) : Thanks for pointing out what we missed. We added the individual points in the revised figures. We drew new graphs using RStudio, and individual data were superimposed on a boxplot applied a scattered function. We hope that these figures would help clear interpretation of our data.

5) In the survival curves, authors must include the number of "patients at risk" for each time interval. This improves readers' interpretation.
Answers to 5) : We agree the reviewer’s opinion. We additionally described patients at risk in the revised figures of survival curves.  

Reviewer 2 Report

The authors present immunohistochemical data and genomic data in a relatively small number of patients regarding the association of apoptotic and proliferative activity with progression from normal hematopoiesis to low-risk MDS. This is not a new finding, as Raza et al first described the relationship between apoptosis and MDS in the early 1990's.  The relevance of this study might benefit from additional longitudinal studies in individual patients over time that examine changes in apoptosis and proliferation that might occur with transformation to high-risk MDS and AML. If such changes are reproducible, then serial measurements might be prognostic/predictive for clonal progression and leukemic transformation.

Specific comments:

1.  Perhaps I missed it, but I did not find any description of methods used to determine telomere length.

2.  Page 3, lines 118-120 state that high cleaved caspase-3 was associated with male sex and more thrombocytopenia, but that statement is not supported by the data presented in Table 1.

3.  A clear definition of "higher risk" MDS would be useful, particularly in terms of marrow blast percentages and genomics.

Author Response

  1. Perhaps I missed it, but I did not find any description of methods used to determine telomere length.

Answer to 1) : We did analysis the telomere length in the previous studies. So, we omitted the methods in this article. Thank you for your advise and we additionally described the methods used to determine telomere length.

Page 3, Lines 95-100 :

2.3 Telomere Q-FISH

Cryopreserved samples were used for telomere analysis by quantitative fluorescence in situ hybridization (Q-FISH). Q-FISH was performed using a Cy3-labeled telomere pep-tide nucleic acid (PNA) FISH kit (DakoCytomation Denmark A/S, Glostrup, Denmark) and a fluorescein isothiocyanate (FITC)-labeled PNA probe for the centromere of chromosome 2 (kindly provided by DakoCytomation).

  1. Page 3, lines 118-120 state that high cleaved caspase-3 was associated with male sex and more thrombocytopenia, but that statement is not supported by the data presented in Table 1.

Answer to 2) : Thank you for your delicate instructions. We made the mistake of incorrectly describing low as high. Sorry for confusing understanding with a mistake, and we corrected the paragraph as below.

Page 3, Lines 126-128 : The high cleaved caspase-3 level group had more female patients and showed more less thrombocytopenia than the low cleaved caspase-3 level group (Table 1).

  1. A clear definition of "higher risk" MDS would be useful, particularly in terms of marrow blast percentages and genomics.

Answer to 3) : We defined  patients with higher risk MDS as patients with high risk and very high risk according to IPSS-R risk stratification system. Patients with higher risk MDS was 31 (40.8%) and patients with lower risk MDS was 35 (59.2%). The ratio of intermediate risk varies depending on the age of onset and it was not clearly determined of dividing into 2 groups in previous studies [2].

And as you mentioned, we additionally analyzed the association between higher risk MDS and marrow blast percentages or genomics. Patients with higher risk MDS were related to high blast percentage in the bone marrow at the diagnosis. And Patients with higher risk MDS statistically have more mutation with Del(5q)/-5 and Del(7q)/-7.

Round 2

Reviewer 2 Report

The authors have addressed the major issues raised during review.